# A Fast, Universal Algorithm
# to Learn Parametric Nonlinear Embeddings

**Miguel Á. Carreira-Perpiñán**
EECS, University of California, Merced
http://eecs.ucmerced.edu

**Max Vladymyrov**
UC Merced and Yahoo Labs
maxv@yahoo-inc.com

## Abstract

Nonlinear embedding algorithms such as stochastic neighbor embedding do dimensionality reduction by optimizing an objective function involving similarities between pairs of input patterns. The result is a low-dimensional projection of each input pattern. A common way to define an out-of-sample mapping is to optimize the objective directly over a parametric mapping of the inputs, such as a neural net. This can be done using the chain rule and a nonlinear optimizer, but is very slow, because the objective involves a quadratic number of terms each dependent on the entire mapping's parameters. Using the method of auxiliary coordinates, we derive a training algorithm that works by alternating steps that train an auxiliary embedding with steps that train the mapping. This has two advantages: 1) The algorithm is universal in that a specific learning algorithm for any choice of embedding and mapping can be constructed by simply reusing existing algorithms for the embedding and for the mapping. A user can then try possible mappings and embeddings with less effort. 2) The algorithm is fast, and it can reuse $N$-body methods developed for nonlinear embeddings, yielding linear-time iterations.

## 1 Introduction

Given a high-dimensional dataset $\mathbf{Y}_{D \times N} = (\mathbf{y}_1, \dots, \mathbf{y}_N)$ of $N$ points in $\mathbb{R}^D$, nonlinear embedding algorithms seek to find low-dimensional projections $\mathbf{X}_{L \times N} = (\mathbf{x}_1, \dots, \mathbf{x}_N)$ with $L < D$ by optimizing an objective function $E(\mathbf{X})$ constructed using an $N \times N$ matrix of similarities $\mathbf{W} = (w_{nm})$ between pairs of input patterns $(\mathbf{y}_n, \mathbf{y}_m)$. For example, the elastic embedding (EE) [5] optimizes:

$$E(\mathbf{X}) = \sum_{n,m=1}^N w_{nm} \left\| \mathbf{x}_n - \mathbf{x}_m \right\|^2 + \lambda \sum_{n,m=1}^N \exp\left( - \left\| \mathbf{x}_n - \mathbf{x}_m \right\|^2 \right) \qquad \lambda > 0. \tag{1}$$

Here, the first term encourages projecting similar patterns near each other, while the second term repels all pairs of projections. Other algorithms of this type are stochastic neighbor embedding (SNE) [15], $t$-SNE [27], neighbor retrieval visualizer (NeRV) [28] or the Sammon mapping [23], as well as spectral methods such as metric multidimensional scaling and Laplacian eigenmaps [2] (though our focus is on nonlinear objectives). Nonlinear embeddings can produce visualizations of high-dimensional data that display structure such as manifolds or clustering, and have been used for exploratory purposes and other applications in machine learning and beyond.

Optimizing nonlinear embeddings is difficult for three reasons: there are many parameters ($NL$); the objective is very nonconvex, so gradient descent and other methods require many iterations; and it involves $\mathcal{O}(N^2)$ terms, so evaluating the gradient is very slow. Major progress in these problems has been achieved in recent years. For the second problem, the spectral direction [29] is constructed by "bending" the gradient using the curvature of the quadratic part of the objective (for EE, this is the graph Laplacian $\mathbf{L}$ of $\mathbf{W}$). This significantly reduces the number of iterations, while evaluating the direction itself is about as costly as evaluating the gradient. For the third problem, $N$-body methods such as tree methods [1] and fast multipole methods [11] approximate the gradient in $\mathcal{O}(N \log N)$

and $\mathcal{O}(N)$ for small dimensions $L$, respectively, and have allowed to scale up embeddings to millions of patterns [26, 31, 34].

Another issue that arises with nonlinear embeddings is that they do not define an "out-of-sample" mapping $\mathbf{F}$: $\mathbb{R}^D \to \mathbb{R}^L$ that can be used to project patterns not in the training set. There are two basic approaches to define an out-of-sample mapping for a given embedding. The first one is a variational argument, originally put forward for Laplacian eigenmaps [6] and also applied to the elastic embedding [5]. The idea is to optimize the embedding objective for a dataset consisting of the $N$ training points and one test point, but keeping the training projections fixed. Essentially, this constructs a nonparametric mapping implicitly defined by the training points $\mathbf{Y}$ and its projections $\mathbf{X}$, without introducing any assumptions. The mapping comes out in closed form for Laplacian eigenmaps (a Nadaraya-Watson estimator) but not in general (e.g. for EE), in which case it needs a numerical optimization. In either case, evaluating the mapping for a test point is $\mathcal{O}(ND)$, which is slow and does not scale. (For spectral methods one can also use the Nyström formula [3], but it does not apply to nonlinear embeddings, and is still $\mathcal{O}(ND)$ at test time.) The second approach is to use a mapping $\mathbf{F}$ belonging to a parametric family $\mathcal{F}$ of mappings (e.g. linear or neural net), which is fast at test time. Directly fitting $\mathbf{F}$ to $(\mathbf{Y}, \mathbf{X})$ is inelegant, since $\mathbf{F}$ is unrelated to the embedding, and may not work well if the mapping cannot model well the data (e.g. if $\mathbf{F}$ is linear). A better way is to involve $\mathbf{F}$ in the learning from the beginning, by replacing $\mathbf{x}_n$ with $\mathbf{F}(\mathbf{y}_n)$ in the embedding objective function and optimizing it over the parameters of $\mathbf{F}$. For example, for the elastic embedding of (1) this means

$$P(\mathbf{F}) = \sum_{n,m=1}^{N} w_{nm} \left\| \mathbf{F}(\mathbf{y}_n) - \mathbf{F}(\mathbf{y}_m) \right\|^2 + \lambda \sum_{n,m=1}^{N} \exp\left( - \left\| \mathbf{F}(\mathbf{y}_n) - \mathbf{F}(\mathbf{y}_m) \right\|^2 \right). \quad (2)$$

This will give better results because the only embeddings that are allowed are those that are realizable by a mapping $\mathbf{F}$ in the family $\mathcal{F}$ considered. Hence, the optimal $\mathbf{F}$ will exactly match the embedding, which is still trying to optimize the objective $E(\mathbf{X})$. This provides an intermediate solution between the nonparametric mapping described above, which is slow at test time, and the direct fit of a parametric mapping to the embedding, which is suboptimal. We will focus on this approach, which we call *parametric embedding (PE)*, following previous work [25].

A long history of PEs exists, using unsupervised [14, 16–18, 24, 25, 32] or supervised [4, 9, 10, 13, 20, 22] embedding objectives, and using linear or nonlinear mappings (e.g. neural nets). Each of these papers develops a specialized algorithm to learn the particular PE they define (= embedding objective and mapping family). Besides, PEs have also been used as regularization terms in semisupervised classification, regression or deep learning [33].

Our focus in this paper is on *optimizing an unsupervised parametric embedding defined by a given embedding objective $E(\mathbf{X})$, such as EE or $t$-SNE, and a given family for the mapping $\mathbf{F}$, such as linear or a neural net*. The straightforward approach, used in all papers cited above, is to derive a training algorithm by applying the chain rule to compute gradients over the parameters of $\mathbf{F}$ and feeding them to a nonlinear optimizer (usually gradient descent or conjugate gradients). This has three problems. First, a new gradient and optimization algorithm must be developed and coded *for each choice of $E$ and $\mathbf{F}$*. For a user who wants to try different choices on a given dataset, this is very inconvenient—and the power of nonlinear embeddings and unsupervised methods in general is precisely as exploratory techniques to understand the structure in data, so a user needs to be able to try multiple techniques. Ideally, the user should simply be able to plug different mappings $\mathbf{F}$ into any embedding objective $E$, with minimal development work. Second, computing the gradient involves $\mathcal{O}(N^2)$ terms each depending on the entire mapping's parameters, which is very slow. Third, both $E$ and $\mathbf{F}$ must be differentiable for the chain rule to apply.

Here, we propose a new approach to optimizing parametric embeddings, based on the recently introduced *method of auxiliary coordinates (MAC)* [7, 8], that partially alleviates these problems. The idea is to solve an equivalent, constrained problem by introducing new variables (the auxiliary coordinates). Alternating optimization over the coordinates and the mapping's parameters results in a step that trains an auxiliary embedding with a "regularization" term, and a step that trains the mapping by solving a regression, both of which can be solved by existing algorithms. Section 2 introduces important concepts and describes the chain-rule based optimization of parametric embeddings, section 3 applies MAC to parametric embeddings, and section 4 shows with different combinations of embeddings and mappings that the resulting algorithm is very easy to construct, including use of $N$-body methods, and is faster than the chain-rule based optimization.

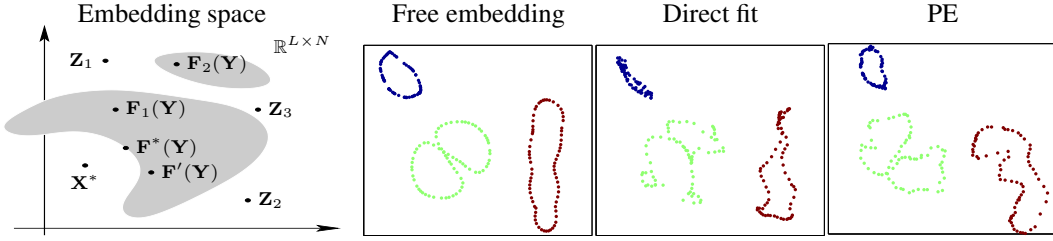

Figure 1: *Left*: illustration of the feasible set $\{\mathbf{Z} \in \mathbb{R}^{L \times N} \colon \mathbf{Z} = \mathbf{F}(\mathbf{Y})$ for $\mathbf{F} \in \mathcal{F}\}$ (grayed areas) of embeddings that can be produced by the mapping family $\mathcal{F}$. This corresponds to the feasible set of the equality constraints in the MAC-constrained problem (4). A parametric embedding $\mathbf{Z}^* = \mathbf{F}^*(\mathbf{Y})$ is a feasible embedding with locally minimal value of $E$. A free embedding $\mathbf{X}^*$ is a minimizer of $E$ and is usually not feasible. A direct fit $\mathbf{F}'$ (to the free embedding $\mathbf{X}^*$) is feasible but usually not optimal. *Right 3 panels*: 2D embeddings of 3 objects from the COIL-20 dataset using a linear mapping: a free embedding, its direct fit, and the parametric embedding (PE) optimized with MAC.

## 2 Free embeddings, parametric embeddings and chain-rule gradients

Consider a given nonlinear embedding objective function $E(\mathbf{X})$ that takes an argument $\mathbf{X} \in \mathbb{R}^{L \times N}$ and maps it to a real value. $E(\mathbf{X})$ is constructed for a dataset $\mathbf{Y} \in \mathbb{R}^{D \times N}$ according to a particular embedding model. We will use as running example the equations (1), (2) for the elastic embedding, which are simpler than for most other embeddings. We call *free embedding* $\mathbf{X}^*$ the result of optimizing $E$, i.e., a (local) optimizer of $E$. A *parametric embedding* (PE) objective function for $E$ using a family $\mathcal{F}$ of mappings $\mathbf{F} \colon \mathbb{R}^D \to \mathbb{R}^L$ (for example, linear mappings), is defined as $P(\mathbf{F}) = E(\mathbf{F}(\mathbf{Y}))$, where $F(\mathbf{Y}) = (\mathbf{F}(\mathbf{y}_1), \ldots, \mathbf{F}(\mathbf{y}_N))$, as in eq. (2) for EE. Note that, to simplify the notation, we do not write explicitly the parameters of $\mathbf{F}$. Thus, a specific PE can be defined by any combination of embedding objective function $E$ (EE, SNE...) and parametric mapping family $\mathcal{F}$ (linear, neural net...). The result of optimizing $P$, i.e., a (local) optimizer of $P$, is a mapping $\mathbf{F}^*$ which we can apply to any input $\mathbf{y} \in \mathbb{R}^D$, not necessarily from among the training patterns. Finally, we call *direct fit* the mapping resulting from fitting $\mathbf{F}$ to $(\mathbf{Y}, \mathbf{X}^*)$ by least-squares regression, i.e., to map the input patterns to a free embedding. We have the following results.

**Theorem 2.1.** *Let $\mathbf{X}^*$ be a global minimizer of $E$. Then $\forall \mathbf{F} \in \mathcal{F} \colon P(\mathbf{F}) \geq E(\mathbf{X}^*)$.*

*Proof.* $P(\mathbf{F}) = E(\mathbf{F}(\mathbf{Y})) \geq E(\mathbf{X}^*)$. ☐

**Theorem 2.2** (Perfect direct fit). *Let $\mathbf{F}^* \in \mathcal{F}$. If $\mathbf{F}^*(\mathbf{Y}) = \mathbf{X}^*$ and $\mathbf{X}^*$ is a global minimizer of $E$ then $\mathbf{F}^*$ is a global minimizer of $P$.*

*Proof.* Let $\mathbf{F} \in \mathcal{F}$ with $\mathbf{F} \neq \mathbf{F}^*$. Then $P(\mathbf{F}) = E(\mathbf{F}(\mathbf{Y})) \geq E(\mathbf{X}^*) = E(\mathbf{F}^*(\mathbf{Y})) = P(\mathbf{F}^*)$. ☐

Theorem 2.2 means that if the direct fit of $\mathbf{F}^*$ to $(\mathbf{Y}, \mathbf{X}^*)$ has zero error, i.e., $\mathbf{F}^*(\mathbf{Y}) = \mathbf{X}^*$, then it is the solution of the parametric embedding, and there is no need to optimize $P$. Theorem 2.1 means that a PE cannot do better than a free embedding[1]. This is obvious in that a PE is not free but constrained to use only embeddings that can be produced by a mapping in $\mathcal{F}$, as illustrated in fig. 1. A PE will typically worsen the free embedding: more powerful mapping families, such as neural nets, will distort the embedding less than more restricted families, such as linear mappings. In this sense, the free embedding can be seen as using as mapping family $\mathcal{F}$ a table $(\mathbf{Y}, \mathbf{X})$ with parameters $\mathbf{X}$. It represents the most flexible mapping, since every projection $\mathbf{x}_n$ is a free parameter, but it can only be applied to patterns in the training set $\mathbf{Y}$. We will assume that the direct fit has a positive error, i.e., the direct fit is not perfect, so that optimizing $P$ is necessary.

Computationally, the complexity of the gradient of $P(\mathbf{F})$ appears to be $\mathcal{O}(N^2 |\mathbf{F}|)$, where $|\mathbf{F}|$ is the number of parameters in $\mathbf{F}$, because $P(\mathbf{F})$ involves $\mathcal{O}(N^2)$ terms, each dependent on all the parameters of $\mathbf{F}$ (e.g. for linear $\mathbf{F}$ this would cost $\mathcal{O}(N^2 LD)$). However, *if manually simplified and coded*, the gradient can actually be computed in $\mathcal{O}(N^2 L + N |\mathbf{F}|)$. For example, for the elastic embedding with a linear mapping $\mathbf{F}(\mathbf{y}) = \mathbf{A}\mathbf{y}$ where $\mathbf{A}$ is of $L \times D$, the gradient of eq. (2) is:

$$\frac{\partial P}{\partial \mathbf{A}} = 2 \sum_{n,m=1}^{N} \left[ \left( w_{nm} - \lambda \exp\left( -\|\mathbf{A}\mathbf{y}_n - \mathbf{A}\mathbf{y}_m\|^2 \right) \right) \times (\mathbf{A}\mathbf{y}_n - \mathbf{A}\mathbf{y}_m)(\mathbf{y}_n - \mathbf{y}_m)^T \right] \quad (3)$$

and this can be computed in $\mathcal{O}(N^2 L + NDL)$ if we precompute $\mathbf{X} = \mathbf{AY}$ and take common factors of the summation over $\mathbf{x}_n$ and $\mathbf{x}_m$. An automatic differentiation package may or may not be able to realize these savings in general.

The obvious way to optimize $P(\mathbf{F})$ is to compute the gradient wrt the parameters of $\mathbf{F}$ by applying the chain rule (since $P$ is a function of $E$ and this is a function of the parameters of $\mathbf{F}$), assuming $E$ and $\mathbf{F}$ are differentiable. While perfectly doable in theory, in practice this has several problems. (1) Deriving, debugging and coding the gradient of $P$ for a nonlinear $\mathbf{F}$ is cumbersome. One could use automatic differentiation [12], but current packages can result in inefficient, non-simplified gradients in time and memory, and are not in widespread use in machine learning. Also, combining autodiff with $N$-body methods seems difficult, because the latter require spatial data structures that are effective for points in low dimension (no more than 3 as far as we know) and depend on the actual point values. (2) The PE gradient may not benefit from special-purpose algorithms developed for embeddings. For example, the spectral direction method [29] relies on special properties of the free embedding Hessian which do not apply to the PE Hessian. (3) Given the gradient, one then has to choose and possibly adapt a suitable nonlinear optimization method and set its parameters (line search parameters, etc.) so that convergence is assured and the resulting algorithm is efficient. Simple choices such as gradient descent or conjugate gradients are usually not efficient, and developing a good algorithm is a research problem in itself (as evidenced by the many papers that study specific combinations of embedding objective and parametric mapping). (4) Even having done all this, the resulting algorithm will still be very slow because of the complexity of computing the gradient: $\mathcal{O}(N^2 L + N |\mathbf{F}|)$. It may be possible to approximate the gradient using $N$-body methods, but again this would involve significant development effort. (5) As noted earlier, the chain rule only applies if both $E$ and $\mathbf{F}$ are differentiable. Finally, all of the above needs to be redone if we change the mapping (e.g. from a neural net to a RBF network) or the embedding (e.g. from EE to $t$-SNE). We now show how these problems can be addressed by using a different approach to the optimization.

## 3 Optimizing a parametric embedding using auxiliary coordinates

The PE objective function, e.g. (2), can be seen as a nested function where we first apply $\mathbf{F}$ and then $E$. A recently proposed strategy, the *method of auxiliary coordinates (MAC)* [7, 8], can be used to derive optimization algorithms for such nested systems. We write the nested problem $\min P(\mathbf{F}) = E(\mathbf{F}(\mathbf{Y}))$ as the following, equivalent constrained optimization problem:

$$\min \bar{P}(\mathbf{F}, \mathbf{Z}) = E(\mathbf{Z}) \qquad \text{s.t.} \qquad \mathbf{z}_n = \mathbf{F}(\mathbf{y}_n), \; n = 1, \dots, N \qquad (4)$$

where we have introduced an auxiliary coordinate $\mathbf{z}_n$ for each input pattern and a corresponding equality constraint. $\mathbf{z}_n$ can be seen as the output of $\mathbf{F}$ (i.e., the low-dimensional projection) for $\mathbf{x}_n$. The optimization is now on an augmented space $(\mathbf{F}, \mathbf{Z})$ with $NL$ extra parameters $\mathbf{Z} \in \mathbb{R}^{L \times N}$, and $\mathbf{F} \in \mathcal{F}$. The feasible set of the equality constraints is shown in fig. 1. We solve the constrained problem (4) using a quadratic-penalty method (it is also possible to use the augmented Lagrangian method), by optimizing the following unconstrained problem and driving $\mu \to \infty$:

$$\min P_Q(\mathbf{F}, \mathbf{Z}; \mu) = E(\mathbf{Z}) + \tfrac{\mu}{2} \sum_{n=1}^{N} \|\mathbf{z}_n - \mathbf{F}(\mathbf{y}_n)\|^2 = E(\mathbf{Z}) + \tfrac{\mu}{2} \|\mathbf{Z} - \mathbf{F}(\mathbf{Y})\|^2. \qquad (5)$$

Under mild assumptions, the minima $(\mathbf{Z}^*(\mu), \mathbf{F}^*(\mu))$ trace a continuous path that converges to a local optimum of $\bar{P}(\mathbf{F}, \mathbf{Z})$ and hence of $P(\mathbf{F})$ [7, 8]. Finally, we optimize $P_Q$ using alternating optimization over the coordinates and the mapping. This results in two steps:

**Over F given Z:** $\min_{\mathbf{F} \in \mathcal{F}} \sum_{n=1}^{N} \|\mathbf{z}_n - \mathbf{F}(\mathbf{y}_n)\|^2$. This is a *standard least-squares regression* for a dataset $(\mathbf{Y}, \mathbf{Z})$ using $\mathbf{F}$, and can be solved using existing, well-developed code for many families of mappings. For example, for a linear mapping $\mathbf{F}(\mathbf{y}) = \mathbf{Ay}$ we solve a linear system $\mathbf{A} = \mathbf{ZY}^+$ (efficiently done by caching $\mathbf{Y}^+$ in the first iteration and doing a matrix multiplication in subsequent iterations); for a deep net, we can use stochastic gradient descent with pretraining, possibly on a GPU; for a regression tree or forest, we can use any tree-growing algorithm; etc. Also, note that if we want to have a regularization term $R(\mathbf{F})$ in the PE objective (e.g. for weight decay, or for model complexity), that term will appear in the $\mathbf{F}$ step but not in the $\mathbf{Z}$ step. Hence, the training and regularization of the mapping $\mathbf{F}$ is confined to the $\mathbf{F}$ step, given the inputs $\mathbf{Y}$ and current outputs $\mathbf{Z}$. The mapping $\mathbf{F}$ "communicates" with the embedding objective precisely through these low-dimensional coordinates $\mathbf{Z}$.

**Over Z given F:** $\min_{\mathbf{Z}} E(\mathbf{Z}) + \frac{\mu}{2} \|\mathbf{Z} - \mathbf{F}(\mathbf{Y})\|^2$. This is a *regularized embedding*, since $E(\mathbf{Z})$ is the original embedding objective function and $\|\mathbf{Z} - \mathbf{F}(\mathbf{Y})\|^2$ is a quadratic regularization term on $\mathbf{Z}$, with weight $\frac{\mu}{2}$, which encourages $\mathbf{Z}$ to be close to a given embedding $\mathbf{F}(\mathbf{Y})$. We can reuse existing, well-developed code to learn the embedding $E(\mathbf{Z})$ with simple modifications. For example, the gradient has an added term $\mu(\mathbf{Z} - \mathbf{F}(\mathbf{Y}))$; the spectral direction now uses a curvature matrix $\mathbf{L} + \frac{\mu}{2}\mathbf{I}$. The embedding "communicates" with the mapping $\mathbf{F}$ through the outputs $\mathbf{F}(\mathbf{Y})$ (which are constant within the $\mathbf{Z}$ step), which gradually force the embedding $\mathbf{Z}$ to agree with the output of a member of the family of mappings $\mathcal{F}$.

Hence, the intricacies of nonlinear optimization (line search, method parameters, etc.) remain confined within the regression for $\mathbf{F}$ and within the embedding for $\mathbf{Z}$, *separately from each other*. Designing an optimization algorithm for an arbitrary combination of embedding and mapping is simply achieved by alternately calling existing algorithms for the embedding and for the mapping.

Although we have introduced a large number of new parameters to optimize over, the $NL$ auxiliary coordinates $\mathbf{Z}$, the cost of a MAC iteration is actually the same (asymptotically) as the cost of computing the PE gradient, i.e., $\mathcal{O}(N^2 L + N|\mathbf{F}|)$, where $|\mathbf{F}|$ is the number of parameters in $\mathbf{F}$. In the $\mathbf{Z}$ step, the objective function has $\mathcal{O}(N^2)$ terms but each term depends only on 2 projections ($\mathbf{z}_n$ and $\mathbf{z}_m$, i.e., $2L$ parameters), hence it costs $\mathcal{O}(N^2 L)$. In the $\mathbf{F}$ step, the objective function has $N$ terms, each depending on the entire mapping's parameters, hence it costs $\mathcal{O}(N|\mathbf{F}|)$.

Another advantage of MAC is that, because it does not use chain-rule gradients, it is even possible to use something like a regression tree for $\mathbf{F}$, which is not differentiable, and so the PE objective function is not differentiable either. In MAC, we can use an algorithm to train regression trees within the $\mathbf{F}$ step using as data $(\mathbf{Y}, \mathbf{Z})$, reducing the constraint error $\|\mathbf{Z} - \mathbf{F}(\mathbf{Y})\|^2$ and the PE objective.

A final advantage is that we can benefit from recent work done on using $N$-body methods to reduce the $\mathcal{O}(N^2)$ complexity of computing the embedding gradient exactly to $\mathcal{O}(N \log N)$ (using tree-based methods such as the Barnes-Hut algorithm; [26, 34]) or even $\mathcal{O}(N)$ (using fast multipole methods; [31]), at a small approximation error. We can reuse such code as is, without any extra work, to approximate the gradient of $E(\mathbf{Z})$ and then add to it the exact gradient of the regularization term $\|\mathbf{Z} - \mathbf{F}(\mathbf{Y})\|^2$, which is already linear in $N$. Hence, each MAC iteration ($\mathbf{Z}$ and $\mathbf{F}$ steps) runs in linear time on the sample size, and is thus scalable to larger datasets.

The problem of optimizing parametric embeddings is closely related to that of learning binary hashing for fast information retrieval using affinity-based loss functions [21]. The only difference is that in binary hashing the mapping $\mathbf{F}$ (an $L$-bit hash function) maps a $D$-dimensional vector $\mathbf{y} \in \mathbb{R}^D$ to an $L$-dimensional binary vector $\mathbf{z} \in \{0,1\}^L$. The MAC framework can also be applied, and the resulting algorithm alternates an $\mathbf{F}$ step that fits a classifier for each bit of the hash function, and a $\mathbf{Z}$ step that optimizes a regularized binary embedding using combinatorial optimization.

**Schedule of $\mu$, initial Z and the path to a minimizer**   The MAC algorithm for parametric embeddings introduces no new optimization parameters except for the penalty parameter $\mu$. The convergence theory of quadratic-penalty methods and MAC [7, 8, 19] tells us that convergence to a local optimum is guaranteed if each iteration achieves sufficient decrease (always possible by running enough $(\mathbf{Z}, \mathbf{F})$ steps) and if $\mu \to \infty$. The latter condition ensures the equality constraints are eventually satisfied. Mathematically, the minima $(\mathbf{Z}^*(\mu), \mathbf{F}^*(\mu))$ of $P_Q$ as a function of $\mu \in [0, \infty)$ trace a continuous path in the $(\mathbf{Z}, \mathbf{F})$ space that ends at a local minimum of the constrained problem (4) and thus of the parametric embedding objective function. Hence, our algorithm belongs to the family of path-following methods, such as quadratic penalty, augmented Lagrangian, homotopy and interior-point methods, widely regarded as effective with nonconvex problems.

In practice, one follows that path loosely, i.e., doing fast, inexact steps on $\mathbf{Z}$ and $\mathbf{F}$ for the current value of $\mu$ and then increasing $\mu$. How fast to increase $\mu$ does depend on the particular problem; typically, one multiplies $\mu$ times a factor of around 2. Increasing $\mu$ very slowly will follow the path more closely, but the runtime will increase. Since $\mu$ does not appear in the $\mathbf{F}$ step, increasing $\mu$ is best done within a $\mathbf{Z}$ step (i.e., we run several iterations over $\mathbf{Z}$, increase $\mu$, run several iterations over $\mathbf{Z}$, and then do an $\mathbf{F}$ step).

The starting point of the path is $\mu \to 0^+$. Here, the $\mathbf{Z}$ step simply optimizes $E(\mathbf{Z})$ and hence gives us a free embedding (e.g. we just train an elastic embedding model on the dataset). The $\mathbf{F}$ step then fits $\mathbf{F}$ to $(\mathbf{Y}, \mathbf{Z})$ and hence gives us the direct fit (which generally will have a positive error

$\|\mathbf{Z} - \mathbf{F}(\mathbf{Y})\|^2$, otherwise we stop with an optimal PE). Thus, *the beginning of the path is the direct fit to the free embedding*. As $\mu$ increases, we follow the path $(\mathbf{Z}^*(\mu), \mathbf{F}^*(\mu))$, and as $\mu \to \infty$, $\mathbf{F}$ converges to a minimizer of the PE and $\mathbf{Z}$ converges to $\mathbf{F}(\mathbf{Y})$. Hence, the "lifetime" of the MAC algorithm over the "time" $\mu$ starts with a free embedding and a direct fit which disagree with each other, and progressively reduces the error in the $\mathbf{F}$ fit by increasing the error in the $\mathbf{Z}$ embedding, until $\mathbf{F}(\mathbf{Y})$ and $\mathbf{Z}$ agree at an optimal PE.

Although it is possible to initialize $\mathbf{Z}$ in a different way (e.g. random) and start with a large value of $\mu$, we find this converges to worse local optima than starting from a free embedding with a small $\mu$. Good local optima for the free embedding itself can be found by homotopy methods as well [5].

## 4   Experiments

Our experiments confirm that MAC finds optima as good as those of the conventional optimization based on chain-rule gradients, but that it is faster (particularly if using $N$-body methods). We demonstrate this with different embedding objectives (the elastic embedding and $t$-SNE) and mappings (linear and neural net). We report on a representative subset of experiments.

**Illustrative example**   The simple example of fig. 1 shows the different embedding types described in the paper. We use the COIL-20 dataset, containing rotation sequences of 20 physical objects every 5 degrees, each a grayscale image of $128 \times 128$ pixels, total $N = 1\,440$ points in $16\,384$ dimensions; thus, each object traces a closed loop in pixel space. We produce 2D embeddings of 3 objects, using the elastic embedding (EE) [5]. The free embedding $\mathbf{X}^*$ results from optimizing the EE objective function (1), without any limitations on the low-dimensional projections. It gives the best visualization of the data, but no out-of-sample mapping. We now seek a linear out-of-sample mapping $\mathbf{F}$. The direct fit fits a linear mapping to map the high-dimensional images $\mathbf{Y}$ to their 2D projections $\mathbf{X}^*$ from the free embedding. The resulting predictions $\mathbf{F}(\mathbf{Y})$ give a quite distorted representation of the data, because a linear mapping cannot realize the free embedding $\mathbf{X}$ with low error. The parametric embedding (PE) finds the linear mapping $\mathbf{F}^*$ that optimizes $P(\mathbf{F})$, which for EE is eq. (2). To optimize the PE, we used MAC (which was faster than gradient descent and conjugate gradients). The resulting PE represents the data worse than the free embedding (since the PE is constrained to produce embeddings that are realizable by a linear mapping), but better than the direct fit, because the PE can search for embeddings that, while being realizable by a linear mapping, produce a lower value of the EE objective function.

The details of the optimization are as follows. We preprocess the data using PCA projecting to 15 dimensions (otherwise learning a mapping would be trivial since there are more degrees of freedom than there are points). The free embedding was optimized using the spectral direction [29] until consecutive iterates differed by a relative error less than $10^{-3}$. We increased $\mu$ from 0.003 to 0.015 with a step of 0.001 (12 $\mu$ values) and did 40 iterations for each $\mu$ value. The $\mathbf{Z}$ step uses the spectral direction, stopping when the relative error is less than $10^{-2}$.

**Cost of the iterations**   Fig. 2(left) shows, as a function of the number of data points $N$ (using a 3D Swissroll dataset), the time needed to compute the gradient of the PE objective (red curve) and the gradient of the MAC $\mathbf{Z}$ and $\mathbf{F}$ steps (black and magenta, respectively, as well as their sum in blue). We use $t$-SNE and a sigmoidal neural net with an architecture 3–100–500–2. We approximate the $\mathbf{Z}$ gradient in $\mathcal{O}(N \log N)$ using the Barnes-Hut method [26, 34]. The log-log plot shows the asymptotically complexity to be quadratic for the PE gradient, but linear for the $\mathbf{F}$ step and $\mathcal{O}(N \log N)$ for the $\mathbf{Z}$ step. The PE gradient runs out of memory for large $N$.

**Quality of the local optima**   For the same Swissroll dataset, fig. 2(right) shows, as a function of the number of data points $N$, the final value of the PE objective function achieved by the chain-rule CG optimization and by MAC, both using the same initialization. There is practically no difference between both optimization algorithms. We sometimes do find they converge to different local optima, as in some of our other experiments.

**Different embedding objectives and mapping families**   The goal of this experiment is to show that we can easily derive a convergent, efficient algorithm for various combinations of embeddings and mappings. We consider as embedding objective functions $E(\mathbf{X})$ $t$-SNE and EE, and as mappings $\mathbf{F}$ a neural net and a linear mapping. We apply each combination to learn a parametric embedding for the MNIST dataset, containing $N = 60\,000$ images of handwritten digit images. Training

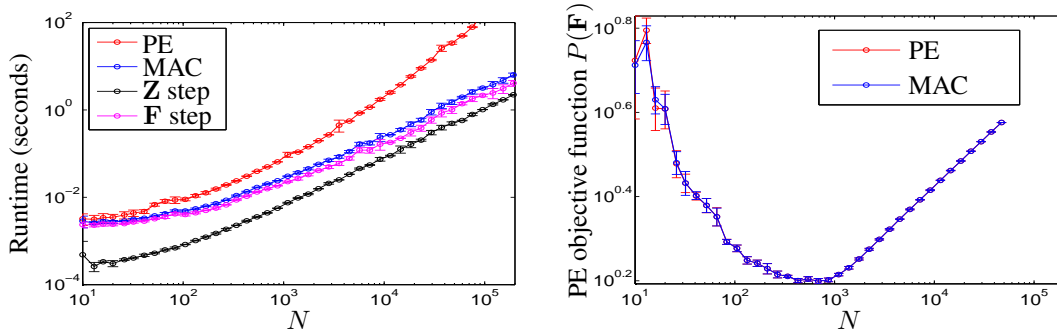

Figure 2: Runtime per iteration and final PE objective for a 3D Swissroll dataset, using as mapping **F** a sigmoidal neural net with an architecture 3–100–500–2, for $t$-SNE. For PE, we give the runtime needed to compute the gradient of the PE objective using CG with chain-rule gradients. For MAC, we give the runtime needed to compute the (**Z**,**F**) steps, separately and together. The gradient of the **Z** step is approximated with an $N$-body method. Errorbars over 5 randomly generated Swissrolls.

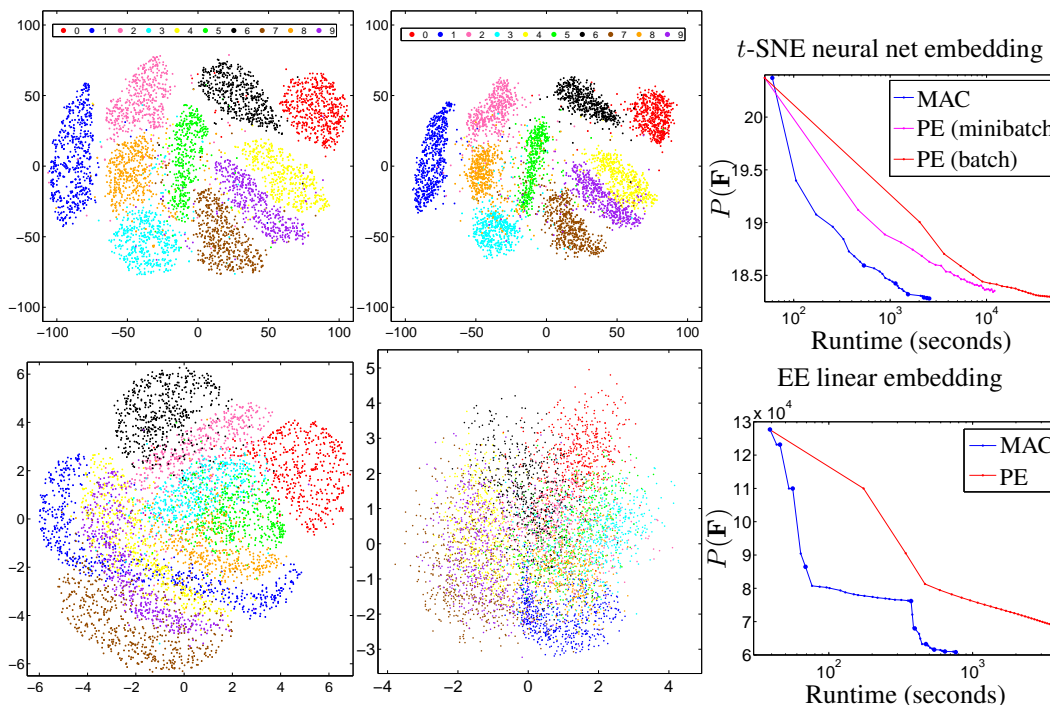

Figure 3: MNIST dataset. *Top*: $t$-SNE with a neural net. *Bottom*: EE with a linear mapping. *Left*: initial, free embedding (we show a sample of 5 000 points to avoid clutter). *Middle*: final parametric embedding. *Right*: learning curves for MAC and chain-rule optimization. Each marker indicates one iteration. For MAC, the solid markers indicate iterations where $\mu$ increased.

a nonlinear (free) embedding on a dataset of this size was very slow until the recent introduction of $N$-body methods for $t$-SNE, EE and other methods [26, 31, 34]. We are the first to use $N$-body methods for PEs, thanks to the decoupling between mapping and embedding introduced by MAC.

For each combination, we derive the MAC algorithm by reusing code available online: for the EE and $t$-SNE (free) embeddings we use the spectral direction [29]; for the $N$-body methods to approximate the embedding objective function gradient we use the fast multipole method for EE [31] and the Barnes-Hut method for $t$-SNE [26, 34]; and for training a deep net we use unsupervised pretraining and backpropagation [22, 25]. Fig. 3(left) shows the free embedding of MNIST obtained with $t$-SNE and EE after 100 iterations of the spectral direction. To compute the Gaussian affinities between pairs of points, we used entropic affinities with perplexity $K = 30$ neighbors [15, 30].

The optimization details are as follows. For the neural net, we replicated the setup of [25]. This uses a neural net with an architecture $(28 \times 28)$–500–500–2000–2, initialized with pretraining as de-

scribed in [22] and [25]. For the chain-rule PE optimization we used the code from [25]. Because of memory limitations, [25] actually solved an approximate version of the PE objective function, where rather than using all $N^2$ pairwise point interactions, only $BN$ interactions are used, corresponding to using minibatches of $B = 5\,000$ points. Therefore, the solution obtained is not a minimizer of the PE objective, as can be seen from the higher objective value in fig. 3(bottom). However, we did also solve the exact objective by using $B = N$ (i.e., one minibatch containing the entire dataset). Each minibatch was trained with 3 CG iterations and a total of 30 epochs.

For MAC, we used $\mu \in \{10^{-7}, 5 \cdot 10^{-7}, 10^{-6}, 5 \cdot 10^{-6}, 10^{-5}, 5 \cdot 10^{-5}\}$, optimizing until the objective function decrease (before the $\mathbf{Z}$ step and after the $\mathbf{F}$ step) was less than a relative error of $10^{-3}$. The rest of the optimization details concern the embedding and neural net, and are based on existing code. The initialization for $\mathbf{Z}$ is the free embedding. The $\mathbf{Z}$ step (like the free embedding) uses the spectral direction with a fixed step size $\gamma = 0.05$, using 10 iterations of linear conjugate gradients to solve the linear system $(\mathbf{L} + \frac{\mu}{2}\mathbf{I})\mathbf{P} = -\mathbf{G}$, and using warm-start (i.e., initialized from the the previous iteration's direction). The gradient $\mathbf{G}$ of the free embedding is approximated in $\mathcal{O}(N \log N)$ using the Barnes-Hut method with accuracy $\theta = 1.5$. Altogether one $\mathbf{Z}$ iteration took around 5 seconds. We exit the $\mathbf{Z}$ step when the relative error between consecutive embeddings is less than $10^{-3}$. For the $\mathbf{F}$ step we used stochastic gradient descent with minibatches of 100 points, step size $10^{-3}$ and momentum rate 0.9, and trained for 5 epochs for the first 3 values of $\mu$ and for 3 epochs for the rest.

For the linear mapping $\mathbf{F}(\mathbf{y}) = \mathbf{A}\mathbf{y}$, we implemented our own chain-rule PE optimizer with gradient descent and backtracking line search for 30 iterations. In MAC, we used 10 $\mu$ values spaced logarithmically from $10^{-2}$ to $10^2$, optimizing at each $\mu$ value until the objective function decrease was less than a relative error of $10^{-4}$. Both the $\mathbf{Z}$ step and the free embedding use the spectral direction with a fixed step size $\gamma = 0.01$. We stop optimizing them when the relative error between consecutive embeddings is less than $10^{-4}$. The gradient is approximated using fast multipole methods with accuracy $p = 6$ (the number of terms in the truncated series). In the $\mathbf{F}$ step, the linear system to find $\mathbf{A}$ was solved using 10 iterations of linear conjugate gradients with warm start.

Fig. 3 shows the final parametric embeddings for MAC, neural-net $t$-SNE (top) and linear EE (bottom), and the learning curves (PE error $P(\mathbf{F}(\mathbf{Y}))$ over iterations). MAC is considerably faster than the chain-rule optimization in all cases.

For the neural-net $t$-SNE, MAC is almost $5\times$ faster than using minibatch (the approximate PE objective) and $20\times$ faster than the exact, batch mode. This is partly thanks to the use of $N$-body methods in the $\mathbf{Z}$ step. The runtimes were (excluding the time taken by pretraining, 40'): MAC: 42'; PE (minibatch): 3.36 h; PE (batch): 15 h; free embedding: 63". Without using $N$-body methods, MAC is $4\times$ faster than PE (batch) and comparable to PE (minibatch). For the linear EE, the runtimes were: MAC: 12.7'; PE: 63'; direct fit: 40".

The neural-net $t$-SNE embedding preserves the overall structure of the free $t$-SNE embedding but both embeddings do differ. For example, the free embedding creates small clumps of points and the neural net, being a continuous mapping, tends to smooth them out. The linear EE embedding distorts the free EE embedding considerably more than if using a neural net. This is because a linear mapping has a much harder time at approximating the complex mapping from the high-dimensional data into 2D that the free embedding implicitly demands.

## 5   Conclusion

In our view, the main advantage of using the method of auxiliary coordinates (MAC) to learn parametric embeddings is that it simplifies the algorithm development. One only needs to plug in existing code for the embedding (with minor modifications) and the mapping. This is particularly useful to benefit from complex, highly optimized code for specific problems, such as the $N$-body methods we used here, or perhaps GPU implementations of deep nets and other machine learning models. In many applications, the efficiency in programming an easy, robust solution is more valuable than the speed of the machine. But, in addition, we find that the MAC algorithm can be quite faster than the chain-rule based optimization of the parametric embedding.

**Acknowledgments**

Work funded by NSF award IIS–1423515. We thank Weiran Wang for help with training the deep net in the MNIST experiment.

## Footnotes

[1]By a continuity argument, theorem 2.2 carries over to the case where $\mathbf{F}^*$ and $\mathbf{X}^* = \mathbf{F}^*(\mathbf{Y})$ are local minimizers of $P$ and $E$, respectively. However, theorem 2.2 would apply only locally, that is, $P(\mathbf{F}) \geq E(\mathbf{X}^*)$ holds locally but there may be mappings $\mathbf{F}$ with $P(\mathbf{F}) < E(\mathbf{X}^*)$ associated with another (lower) local minimizer of $E$. However, the same intuition remains: we cannot expect a PE to improve over a good free embedding.

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
