[Supplementary Material]

# Supplementary material for:
# A Fast, Universal Algorithm
# to Learn Parametric Nonlinear Embeddings

Miguel Á. Carreira-Perpiñán       Max Vladymyrov*
Electrical Engineering and Computer Science, University of California, Merced
`http://eecs.ucmerced.edu`

September 15, 2015

**Abstract**

This contains extended experiments. The equation numbers referred to correspond to the main paper.

## 1   Experiments

Our experiments confirm that MAC finds optima as good as those of the conventional optimization based on chain-rule gradients, but that it is faster (particularly if using $N$-body methods). We demonstrate this with different embedding objectives (the elastic embedding and $t$-SNE) and mappings (linear and neural net).

### 1.1   Illustrative example: free embedding, parametric embedding and direct fit

The simple example of fig. 1 shows the different embedding types described in the paper. We use the COIL-20 dataset [3], containing rotation sequences of 20 physical objects every 5 degrees, each a grayscale image of $128 \times 128$ pixels, total $N = 1\,440$ points in $16\,384$ dimensions; thus, each object traces a closed loop in pixel space. We produce 2D embeddings of 3 objects, using the elastic embedding (EE) [1]. The free embedding $\mathbf{X}^*$ results from optimizing the EE objective function (1), without any limitations on the low-dimensional projections. It gives the best visualization of the data, but no out-of-sample mapping. We now seek a linear out-of-sample mapping $\mathbf{F}$. The direct fit fits a linear mapping to map the high-dimensional images $\mathbf{Y}$ to their 2D projections $\mathbf{X}^*$ from the free embedding. The resulting predictions $\mathbf{F}(\mathbf{Y})$ give a quite distorted representation of the data, because a linear mapping cannot realize the free embedding $\mathbf{X}$ with low error. The parametric embedding (PE) finds the linear mapping $\mathbf{F}^*$ that optimizes $P(\mathbf{F})$, which for EE is eq. (2). To optimize the PE, we used MAC, and two methods based on chain-rule gradients (which we implemented in Matlab): gradient descent (not shown) and conjugate gradients (CG). Since the PE objective has local optima, MAC happens to converge to a different (and better) optimum than CG, and is also faster. Both optima produce an embedding that represents the data worse than the free embedding (since the PE is constrained to produce embeddings that are realizable by a linear mapping), but better than the direct fit, because the PE can search for embeddings that, while being realizable by a linear mapping, produce a lower value of the EE objective function.

The details of the optimization are as follows. For all methods, we preprocess the data using PCA projecting to 15 dimensions (otherwise learning a mapping would be trivial since there are more degrees of freedom than there are points). The free embedding was optimized using the spectral direction [7] until consecutive iterates differed by a relative error less than $10^{-3}$. The PE was optimized using CG or gradient descent for 150 iterations. For MAC we initialized $\mathbf{Z}$ to the free embedding and increased $\mu$ from 0.003 to

Figure 1: *Left*: 2D embeddings of 3 objects from the COIL-20 dataset using a linear mapping: a free embedding $\mathbf{X}^* = \arg\min E(\mathbf{X})$, its direct fit (of $\mathbf{F}$ to $(\mathbf{Y}, \mathbf{X}^*)$), and the parametric embedding (PE) $\mathbf{F}^*(\mathbf{Y})$ optimized with conjugate gradients and MAC. *Right*: PE learning curves for gradient descent, CG and MAC.

0.015 with a step of 0.001 (12 $\mu$ values) and did 40 iterations for each $\mu$ value. The $\mathbf{Z}$ step uses the spectral direction, stopping when the relative error is less than $10^{-2}$.

In this and subsequent learning curves we plot the PE error $P(\mathbf{F}(\mathbf{Y}))$. While MAC minimizes this as $\mu \to \infty$, each MAC iteration need not decrease the PE error monotonically (since it optimizes $P_Q(\mathbf{Z}, \mathbf{F}; \mu)$ instead). However, we rarely observe iterations where the PE error actually increases.

## 1.2 Cost of the iterations

Fig. 2(top) shows, as a function of the number of data points $N$ (using a 3D Swissroll dataset), the time needed to compute the gradient of the PE objective (red curve) and the gradient of the MAC $\mathbf{Z}$ and $\mathbf{F}$ steps (black and magenta, respectively, as well as their sum in blue). As embedding objective $E(\mathbf{X})$ we use EE (top plot) and $t$-SNE (bottom plot) and as mapping $\mathbf{F}$ we use a neural net with an architecture 3–100–500–2 with sigmoidal activations. We approximate the $\mathbf{Z}$ gradient in linear time for EE using a fast multipole method [9] and for $t$-SNE in $\mathcal{O}(N \log N)$ using the Barnes-Hut method [6, 10]. The log-log plot shows asymptotically quadratic complexity for the PE gradient, but linear complexity for the $\mathbf{Z}$ and $\mathbf{F}$ steps ($\mathcal{O}(N \log N)$ for the $\mathbf{Z}$ step in $t$-SNE). The PE gradient runs out of memory for large $N$.

## 1.3 Quality of the local optima

For the same Swissroll dataset, fig. 2(bottom) shows, as a function of the number of data points $N$, the final value of the PE objective function achieved by the chain-rule CG optimization and by MAC, both using the same initialization. Except when using very small training sets there is practically no difference between both optimization algorithms. We sometimes do find they converge to different local optima, as in some of our other experiments.

## 1.4 Different embedding objectives and mapping families

The goal of this experiment is to show that we can easily derive a convergent, efficient algorithm for various combinations of embeddings and mappings. We consider as embedding objective functions $E(\mathbf{X})$ $t$-SNE and EE, and as mappings $\mathbf{F}$ a neural net and a linear mapping. We apply each combination to learn a parametric embedding for the MNIST dataset, containing $N = 60\,000$ images of handwritten digit images. Training a nonlinear (free) embedding on a dataset of this size was very slow until the recent introduction of $N$-body

Figure 2: Runtime per iteration (*top*) and final PE objective value (*bottom*) for a 3D Swissroll dataset, using as mapping **F** a neural net with an architecture 3–100–500–2 with sigmoidal activations, for EE (*left*) and *t*-SNE (*right*). For PE, we use CG optimization with chain-rule gradients and give the runtime needed to compute the gradient of the PE objective. For MAC, we give the runtime needed to compute the (**Z**,**F**) steps, separately and in combination. The gradient of the **Z** step is approximated with an $N$-body method. The errorbars are over 5 randomly generated Swissrolls.

methods for *t*-SNE, EE and other methods [6, 9, 10]. We are the first to use $N$-body methods for PEs, thanks to the decoupling between mapping and embedding introduced by MAC.

For each combination, we derive the MAC algorithm by reusing code available online: for the EE and *t*-SNE (free) embeddings we use the spectral direction [7]; for the $N$-body methods to approximate the embedding objective function gradient we use the fast multipole method for EE [9] and the Barnes-Hut method for *t*-SNE [6, 10]; and for training a deep net we use unsupervised pretraining and backpropagation [4, 5]. Fig. 3 shows the free embedding of MNIST obtained with *t*-SNE and EE after 100 iterations of the spectral direction. To compute the Gaussian affinities between pairs of points, we used entropic affinities with perplexity $K = 30$ neighbors [2, 8].

**$t$-SNE and EE with a neural net** We replicated the setup of van der Maaten [5]. This uses a neural net with an architecture $(28 \times 28)$–500–500–2000–2, initialized with pretraining as described in Salakhutdinov and Hinton [4] and van der Maaten [5].

Figure 3: 2D free embeddings of 60 000 MNIST images with $t$-SNE and EE. In this and all other MNIST embeddings plots, we show a sample of 5 000 points to avoid clutter.

For the chain-rule PE optimization we used the code from van der Maaten [5]. Because of memory limitations, van der Maaten [5] actually solved an approximate version of the PE objective function, where rather than using all $N^2$ pairwise point interactions, only $BN$ interactions are used, corresponding to using minibatches of $B = 5\,000$ points. Therefore, the solution obtained is not a minimizer of the PE objective, as can be seen from the higher objective value in fig. 4(bottom). However, we did also solve the exact objective by using $B = N$ (i.e., one minibatch containing the entire dataset). Each minibatch was trained with 3 CG iterations and a total of 30 epochs.

For MAC, we used $\mu \in \{10^{-7}, 5\cdot10^{-7}, 10^{-6}, 5\cdot10^{-6}, 10^{-5}, 5\cdot10^{-5}\}$, optimizing until the objective function decrease (before the $\mathbf{Z}$ step and after the $\mathbf{F}$ step) was less than a relative error of $10^{-3}$. The rest of the optimization details concern the embedding and neural net, and are based on existing code. The initialization for $\mathbf{Z}$ is the free embedding. The $\mathbf{Z}$ step (like the free embedding) uses the spectral direction with a fixed step size $\gamma = 0.05$, using 10 iterations of linear conjugate gradients to solve the linear system $(\mathbf{L} + \frac{\mu}{2}\mathbf{I})\mathbf{P} = -\mathbf{G}$, and using warm-start (i.e., initialized from the the previous iteration's direction). The gradient $\mathbf{G}$ of the free embedding is approximated in $\mathcal{O}(N \log N)$ using the Barnes-Hut method with accuracy $\theta = 1.5$. Altogether one $\mathbf{Z}$ iteration took around 5 seconds. We exit the $\mathbf{Z}$ step when the relative error between consecutive embeddings is less than $10^{-3}$. For the $\mathbf{F}$ step we used stochastic gradient descent with minibatches of 100 points, step size $10^{-3}$ and momentum rate 0.9, and trained for 5 epochs for the first three values of $\mu$ and for 3 epochs for the rest.

Fig. 4 shows the final $t$-SNE parametric embedding for MAC (left panel) and the learning curves (right panel). MAC is considerably faster than the chain-rule optimization: almost $5\times$ faster than using minibatch (the approximate PE objective) and $20\times$ faster than the exact, batch mode. This is partly thanks to the use of $N$-body methods in the $\mathbf{Z}$ step. The runtimes were (excluding the time taken by pretraining, 40'): MAC: 42'; PE (minibatch): 3.36 h; PE (batch): 15 h; free embedding: 63". Without using $N$-body methods, MAC is $4\times$ faster than PE (batch) and comparable to PE (minibatch).

The neural-net $t$-SNE embedding preserves the overall structure of the free $t$-SNE embedding but both embeddings do differ. For example, the free embedding creates small clumps of points and the neural net, being a continuous mapping, tends to smooth them out.

Figure 4: *Left*: 2D parametric embedding of 60 000 MNIST images using *t*-SNE with a neural net, optimized with MAC. *Right*: learning curves for different optimization algorithms. Each marker indicates one iteration. For MAC, the solid markers indicate iterations where $\mu$ increased.

**t-SNE and EE with a linear mapping**   Now we use a linear mapping $\mathbf{F}(\mathbf{y}) = \mathbf{A}\mathbf{y}$. We used the same affinity matrix and free embedding as with the neural net. In MAC, we used 10 $\mu$ values spaced logarithmically from $10^{-2}$ to $10^2$, optimizing at each $\mu$ value until the objective function decrease was less than a relative error of $10^{-4}$. Both the $\mathbf{Z}$ step and the free embedding use the spectral direction with a fixed step size $\gamma = 0.01$. We stop optimizing them when the relative error between consecutive embeddings is less than $10^{-4}$. The gradient is approximated using fast multipole methods with accuracy $p = 6$ (the number of terms in the truncated series). In the $\mathbf{F}$ step, the linear system to find $\mathbf{A}$ was solved using 10 iterations of linear conjugate gradients with warm start. We implemented our own chain-rule PE optimizer (for EE only) with gradient descent and backtracking line search for 30 iterations.

Fig. 5 shows the results for EE and *t*-SNE. The chain-rule optimization (for EE) was again quite slower than MAC. The runtimes were: MAC: 12.7'; PE: 63'; direct fit: 40". The resulting parametric linear embedding for both EE and *t*-SNE distorts the corresponding free embedding considerably more than if using a neural net. This is because a linear mapping has a much harder time at approximating the complex mapping from the high-dimensional data into 2D that the free embedding implicitly demands. Fig. 6 shows a PCA embedding for comparison. All embeddings (EE, *t*-SNE, PCA) have been rotated and scaled using Procrustes alignment so they can be more easily compared.

## Footnotes

*Now at Yahoo Labs.