[Reviews · NeurIPS 2015]

Submitted by Assigned_Reviewer_1

I found the quality and clarity of the work to be truly excellent.

As a reader I felt that I truly learned something worth remembering by reading the paper, which sadly is not always the case even among papers accepted for publication at NIPS.

The originality of the work is solid, but bounded given the existence of prior work on method of auxiliary coordinates (MAC), though none of that touches this application area that I am aware of.

The significance (or impact) of the work is similarly material, yet circumscribed.

Concretely, I wouldn't be surprised for many researchers in need of a faster method for optimizing non-linear parametric embeddings to consider MAC in the light of this researcher.

For example, I could imagine handy open source implementations of t-SNE dimensionality reduction to switch to MAC.

That's a solid and measurable impact on the community, but obvious limited in it's core scientific impact.
Summary: An exceptionally clear paper describing the application of method of auxiliary coordinates to finding non-linear embeddings (dimensionality reduction).

I enjoyed the paper and felt that it demonstrates a clear improvement over slower gradient based methods; this is however understandably not the highest impact sort of work one could do.

Submitted by Assigned_Reviewer_2

Note 1 - this is a 'light review'. Note 2 - I was not able to open the supplementary file, the provided archive .gz did not contain any readable file type.

The paper presents an optimization approach for parametric out-of-sample extension of nonlinear dimensionality reduction algorithms.

Solutions are constrained a desired parametric family whose parameters are optimized based on an auxiliary coordinate approach, alternatively optimizing a regularized loss with respect to auxiliary coordinates and then the mapping as a regression to the coordinates.

Experiments on three data sets show that the method achieves solutions about equal in quality to direct optimization of parametric embeddings but is faster. Another advantage is that it does not require separate derivation of optimization equations for each loss + parametric family combination.

Evaluation:

The idea is good and out of sample extension is a common problem in nonlinear dimensionality reduction. The experiments are somewhat brief as they are only on COIL, swissroll and MNIST data; the supplementary file was said to contain more experiments but I was unable to open it.

It is hard to evaluate the contribution. On one hand the method answers a clear need. On the other hand, the method seems mostly an adaptation of the principle in ref. 6 from the multilayer network case in that paper to a general loss function setting.

Details:

* It is stated "Directly fitting F to (Y,X) is inelegant, since F is unrelated to the embedding, and may not work well if the mapping cannot model well the data" but then state "Alternating optimization ... trains an auxiliary embedding with a 'regularization' term, and ... trains the mapping by solving a regression". Both approaches use a regressor, the difference is in the use of regularization during the alternating optimization; it would be useful to clarify this.

* "if manually simplified and coded, the gradient can actually be computed in O(N2L + N |F|)": this would seem to require some proof to show this holds in a general case; the elastic embedding is only a special case.

* "Deriving, debugging and coding the gradient of P for a nonlinear F is cumbersome.": how cumbersome is this? If I understand correctly, many nonlinear dimensionality reduction methods are optimized by gradient algorithms anyway, so the gradient with respect to data points is likely known, all that remains is the right-hand term in the chain rule (gradient of coordinates with respect to the mapping). Do you mean that simplification is the cumbersome part?

* "each term depends only on 2 projections (zn and zm", what are zn and zm?

* "it is even possible to use something like a regression tree for F, which is not differentiable, and so the PE objective function is not differentiable either": does such a non-differentiable F still satisfy the convergence conditions mentioned below eq. (5)?

Summary: Useful optimization method for out-of-sample extension. Experiments only on three data sets. Basic idea of the method follows from a previous nested-function-optimization case, but answers a clear need in dimensionality reduction.

Submitted by Assigned_Reviewer_3

The paper could benefit from testing their method on larger datasets such as skipgram embedding vectors or the pen-ultimate layer of convnets where the number of classes or examples is quite large. This would provide a convincing demonstration of this method compared to other methods.
Summary: The authors present a more efficient method or performing nonlinear embeddings of complex data. The authors compare their results to other state of the art methods and find that this embedding method performs comparably while demands less computation time.

Submitted by Assigned_Reviewer_4

This paper presents an accelerated optimization method for parametric nonlinear dimensionality reduction. The proposed method relaxes the nested nonlinear function by an additional quadratic penalty. The new method is tested in two similarity-based nonlinear dimensionality reduction methods t-SNE and EE and on two image data sets, with significantly reduced training times.

The novelty is marginal. The work is an obvious consequence of an existing work (MAC, in Ref.[6]). Such simple Lagrangian relaxation is a well-known technique in the NIPS community and even in classical optimization textbooks. This work simply reuses the same technique in EE- or SNE-like objectives.

Despite the reported speedup, the proposed method requires the user to specify a sequence of mu values, which is inconvenient. The paper only gives a rough guideline that mu should increase during the iterations. In practice, setting the sequence is still difficult because mu can be coupled with other parameters (e.g. the learning step size gamma and the EE parameter lambda). In the presented experiments, the mu and gamma values are data dependent. They vary from data sets and also from learning objectives, which indicates the user must face the tricky tuning problem.

The EE method seems not suitable for the MNIST data. Especially, Figure 3 middle bottom is a mess, which is even worse than a simple PCA.

After feedback: The author feedback is not satisfied. I believe that there is no major innovation and the method is inconvenient and can be unreliable. The linear EE is even worse than simple PCA for MNIST. It is unreasonable to show this result. The authors did not answer my question. I cannot open the supplemental file.

The above problems still remain in the final version.

Summary: An accelerated optimization method that reuses an existing technique. It is fast but not convenient.

Author Feedback
Author rebuttal: We thank all reviewers for their comments.

First we address the issue of novelty. Our algorithm results indeed from applying the general MAC framework proposed in [6] to parametric embeddings (PEs). The vast majority (all?) of the work on accelerating the training of (parametric) nonlinear embeddings relies on applying existing, known techniques, with some adjustment to the specific embedding objective function considered. For example:

[19,22] Conjugate gradients and deep net pretraining
[23,30] Barnes-Hut algorithm
[26] Partial-Hessian
[27] Barnes-Hut and fast multipole method
...and all the references in l. 82ff. Another recent example:
Zhirong Yang et al AISTATS'15: majorization-minimization, Partial-Hessian

It does not seem reasonable to expect a paper in PEs to come up with an optimization method never proposed before for any other problem. Moreover, MAC is a very recent framework, not widely known, and its application to a new functional form does require the solution of the Z step. For PEs, this results in novel insights:
- The original MAC paper focuses on objective functions that are a sum of N unary terms (one per data point and coordinate zn), rather than a sum over (a subset of) N^2 pairwise terms (zn,zm) as with nonlinear embeddings. This means that the Z step does not separate over the N data points, and is far more complex. We solve this by capitalizing on existing "free" embedding algorithms and their accelerations, such as N-body methods, to solve the Z step.
- It leads to the new concepts we introduce of free and regularized embedding and direct fit and their relations (section 2).

But the fundamental advance is practical. Ours is the first PE algorithm that runs in less than O(N^2) time: linear with FMM, O(N*logN) with Barnes-Hut. Together with the ease of developing algorithms for different combinations of embedding and mapping (very useful for exploratory data analysis), this could make MAC a standard approach for PEs. Having read our paper, the applicability and resulting advantages of MAC for PEs may seem obvious, but ours is the first paper to propose this and evaluate it empirically.

Assigned_Reviewer_3

- mu schedules appear in quad-penalty, aug.Lagrangian, homotopy and interior-point methods -- all effective in nonconvex problems. As with learning rate schedules in stochastic gradient descent, this requires trial & error but is quite doable.

- EE not suitable for MNIST (fig3 middle bottom): Fig5 (suppl.mat.) shows the t-SNE linear embedding for MNIST is no better than EE's. The reason is *the mapping is linear*; the PE is much better with a deep net. Regardless, the job of the MAC (or chain-rule) optimization is to train the PE with the choice of mapping/embedding the user gives.

Assigned_Reviewer_4

+ how is MAC different from ADMM? ADMM is a particular case of the augmented Lagrangian method when the objective separates. For ADMM to apply, we first must reformulate the problem to introduce such separability (often through a consensus transformation). MAC can be seen as a special transformation for nested functions. In fact, as we note in l. 204, we could use the augmented Lagrangian instead of the quadratic-penalty method to solve problem (4).

Assigned_Reviewer_5

Our supplementary file is called supplmat.tar.gz and contains 4 files (pdf, README, 2 animations). Running "tar xzvf supplmat.tar.gz" should extract them (maybe renaming the file to .tgz?).

* "Both approaches (direct fit of F to (Y,X) and MAC) use a regressor, the difference is in the use of regularization during the alternating optimization".
Good point, thanks. It also reiterates the fact that the crucial practical difference over the direct fit is that the output coordinates X (or rather Z) are gradually adapted to the family of functions F (in the coordinate step).

* "if manually simplified and coded, the gradient can actually be computed in O(N2L + N|F|)": this would require proof in general; the elastic embedding is only a special case.
Absolutely, and this is further evidence of the advantage of MAC in the PE case. Without simplification, the chain-rule gradient would be O(N^2 * |F|) (l. 156), much more expensive than for our MAC algorithm.

* cumbersome gradient: even without simplification, coding chain-rule gradients with matrix parameters does take work and is prone to bugs. We know by experience...

* "each term depends only on 2 projections (zn and zm":
zn and zm are the low-dimensional projections of points yn and ym, resp. (as defined near eq. 4).

* Non-differentiable F does not satisfy the convergence conditions mentioned below eq. (5), because the KKT conditions (which need gradients) don't apply. The convergence theory here would depend on conditions to learn the function F itself in the regression. Still, assuming we have a way of doing that even approximately (as with regression trees), the algorithm would apply.